**Fractional governing equations of transient groundwater flow in unconfined**
**aquifers with multi-fractional dimensions in fractional time**
**M. Levent Kavvas[1], Tongbi Tu[2,3], Ali Ercan[2], and James Polsinelli[1]**
[1]Hydrologic Research Laboratory, Department of Civil and Environmental Engineering,
University of California, Davis, CA 95616, USA.
[2]J. Amorocho Hydraulics Laboratory (JAHL), Department of Civil and Environmental
Engineering, University of California - Davis, CA, 95616, USA
[3]Now at Department of Environmental Science, Policy and Management, University of California,
Berkeley, CA 94720.
Correspondence to: M. Levent Kavvas (mlkavvas@ucdavis.edu)
**Abstract:** In this study a dimensionally-consistent governing equation of transient unconfined
groundwater flow in fractional time and multi-fractional space is developed. First, a fractional
continuity equation for transient unconfined groundwater flow is developed in fractional time and
space. For the equation of groundwater motion within a multi-fractional multi-dimensional
unconfined aquifer, a previously-developed dimensionally consistent equation for water flux in
unsaturated/saturated porous media is combined with the Dupuit approximation to obtain an
equation for groundwater motion in multi-fractional space in unconfined aquifers. Combining the
fractional continuity and groundwater motion equations, the fractional governing equation of
transient unconfined aquifer flow is then obtained. Finally, two numerical applications to
unconfined aquifer groundwater flow are presented to show the skills of the proposed fractional
governing equation. As shown in one of the numerical applications, the newly-developed
governing equation can produce heavy-tailed recession behavior in unconfined aquifer discharges.
**1. Introduction**

Nearly 70 years ago in his hydrologic studies of the High Aswan Dam, Hurst (1951) has discovered that the flow time series of the Nile river demonstrated fluctuations whose rescaled range may not be proportional to the square root of the observation duration, but may be proportional to the duration raised to a power H (the so-called Hurst coefficient) that is larger than 0.5 but less than 1. This finding, now called as the "Hurst phenomenon" implies that in such river flows the integral scale (the integral of the flow autocorrelation function with respect to the time lag, over the range from zero to infinity) may not exist, putting the process outside the Brownian domain of finite-memory processes where the integral scale is finite. Since the Hurst phenomenon amounts to the clustering of wet years with wet years and the dry years with the dry years, the so-called "Joseph effect" in the Bible (Mandelbrot, 1977), it has important consequences on the planning and operation of water storage systems over long periods (Koutsoyiannis, 2005). Hurst phenomenon in hydrologic flow processes was later demonstrated convincingly by various researchers, including Eltahir (1996), Radziejewski and Kundzewicz (1997), Montanari et al. (1997), and Vogel et. al. (1998) among others. In order to model the Hurst phenomenon in river flows the fractional Gaussian noise (FGN), where the rescaled range for the time series of a flow process in a time interval [0, t] is proportional to $t^H$ for 0.5<H<1, was introduced by Mandelbrot and Wallis (1969). FGN model was later extended by Koutsoyiannis (2002) in order to model satisfactorily a range of time scales, including the conventional Brownian finite memory flow processes. Aside from the FGN models, physically-based models of the Hurst phenomenon were also developed by various authors, including Klemes (1974), Beran (1994) and Koutsoyiannis (2003). However, a physically-based model that explains the Hurst phenomenon explicitly in terms of the hydrologic process mechanisms is still missing. Yevjevich (1963, 1971) provided a plausible physical explanation for the Markovian structure of the annual river flows within a river basin by linking the annual evolution of the water storage in the basin to the exponential recession in baseflow of the basin runoff. Meanwhile, baseflow in basin runoff is mainly due to unconfined aquifer flow to the neighbouring stream network of the basin. As shall be shown in a numerical example later in this paper, the conventional unconfined groundwater flow equation with integer powers does result in the hydraulic head of and the discharge from the aquifer to decay exponentially, that would result in the Markovian finite memory behaviour of the river outflow from the basin. Such exponentially decaying baseflow, while it can be explained by the mechanics

of the conventional unconfined groundwater flow governing equation with integer powers, may
not produce the heavy tailed recession behaviour necessary for the long range dependence in river
flows, the basic characteristic of the Hurst phenomenon, reported in annual river flow series in the
above-mentioned studies. The conventional integer-power governing equations of the unconfined
groundwater flow, having finite memory, are fundamentally in the Brownian domain, and may not
model the heavy-tailed baseflow recession behaviour that would be necessary to model the Hurst
phenomenon in annual river flows.  What is needed is a new structure for the governing equation
of unconfined groundwater flow that can reproduce heavy tailed behaviour with time in the
hydraulic head and aquifer discharge recession, that would then lead to heavy-tailed recession
behaviour in the baseflow of the river basin. Furthermore, various researchers also reported long-
range dependence in groundwater level fluctuations (e.g., Li and Zhang, 2007; Yu et al., 2016; Tu
et al., 2017; and the references therein). One possible way to reproduce heavy-tailed recession
behavior in the hydraulic head and discharge of an unconfined aquifer is by means of a new
governing equation of unconfined groundwater flow with fractional powers. Such behavior in an
anisotropic confined groundwater aquifer with time and space fractional operators in its governing
equation was recently demonstrated (Kavvas et al. 2017a, Tu et al. 2017). Accordingly, the
reported study will follow a similar approach to develop a new governing equation for unconfined
groundwater aquifers.

Reporting  that conventional geometries cannot characterize groundwater flow in many

fractured rock aquifers (Black et al., 1986), and the observed drawdown tends to be underestimated
in early times and overestimated at later times by the conventional radial groundwater flow model
(Van Tonder et al., 2001), Cloot and Botha (2006) developed a fractional governing equation for
radial groundwater flow in integer time and fractional space in a uniform homogeneous aquifer.
They used the Riemann-Liouville (RL) fractional derivative form (please see Podlubny, 1998 page
62-77, for a comprehensive explanation of the RL fractional derivative) in their model formulation.
Atangana and Bildik (2013), Atangana (2014),  and Atangana and Vermeulen (2014) then
reformulated the fractional radial groundwater flow model of Cloot and Botha (2006) by the
Caputo differentiation framework (to be detailed in the next section) , and reported better
performance. Compared to the Riemann-Liouville derivative approach, the Caputo framework has
a fundamental advantage of being able to accommodate physically-interpretable real-life initial
and boundary conditions (Podlubny, 1998). In simple terms, a differential equation which is
based on Riemann-Liouville (RL) fractional derivative, requires the limit values of the RL
fractional derivative for its initial and boundary values which have no known physical
interpretation (Podlubny, 1998, page 78). Meanwhile, "Caputo derivatives take on the same form
as for integer-order differential equations, i.e. contain the limit values of integer-order
derivatives…" (Podlubny, 1998, page 79) incorporating the real world initial and boundary
conditions into the solution of a fractional governing equation.  Atangana and Baleanu (2014)
presented a new radial groundwater flow model in fractional time based on a new fractional
derivative definition, "conformable derivative" (Khalil et al., 2014). Most recently, Su (2017)
proposed a time-space fractional Boussinesq equation and he claimed this fractional equation is a
general groundwater flow equation and can be applied to groundwater flow in both confined and
unconfined aquifers. However, all of the aforementioned studies only presented the formulated
fractional governing groundwater flow equations and no detailed derivations of these governing
equations from the fundamental conservation principles were provided.
Wheatcraft and Meerschaert (2008) derived the groundwater flow continuity equation in the
fractional form by using the fractional Taylor series approximation. They further removed the
linearity / piecewise linearity restriction for the flux and the infinitesimal control volume
restriction. When developing the fractional continuity equation, the groundwater flow process was
considered in fractional space but in integer time by Wheatcraft and Meerschaert (2008). They
further assumed the same fractional power in every direction of the fractional porous media space.
Furthermore, only the mass conservation was considered in their derivation, but not the fractional
water flux equation. Mehdinejadiani et al. (2013) expanded the approach of Wheatcraft and
Meerschaert (2008) to the derivation of a governing equation of groundwater flow in an
unconfined aquifer in fractional space but in integer time. In their derivation, they used the
conventional Darcy formulation for the water flux with integer spatial derivative while utilizing
fractional spatial derivatives in their continuity equation.
Olsen et al. (2016) pointed out that the derivations in Wheatcraft and Meerschaert (2008) and
Mehdinejadiani et al. (2013) utilized the fractional Taylor series, as formulated by Odibat and
Shawagfeh (2007), which utilized local Caputo derivatives. In order to expand the local Caputo
derivatives in the above-mentioned studies, Olsen et al. (2016) utilized the fractional mean value
theorem from Diethelm (2012) to develop a continuity equation of groundwater flow with left and
right fractional nonlocal Caputo derivatives in fractional space but in integer time. Olsen et al.
(2016) did not address the water flux formulation in fractional space, and, hence, did not develop
a complete governing equation of groundwater flow. They also did not address the multifractional
spatial derivatives in order to address anisotropy within an aquifer. Around that time, Kavvas et
al. (2017a) utilized the mean value formulation from Usero (2007), Odibat and Shawagfeh (2007)
and Li et al. (2009) to derive a complete governing equation of transient groundwater flow in a
confined, anisotropic aquifer with fractional time and multi-fractional space derivatives which
addressed not only the continuity but also the water flux (motion) in fractional time-space and the
effect of a sink/source term. By employing the above-mentioned fractional mean value
formulations, Kavvas et al. (2017a) developed the governing equation of confined groundwater
flow in fractional time-space in non-local form.
As mentioned above, unconfined groundwater flow is the fundamental component of the
watershed runoff baseflow since it is the fundamental contributor to the network streamflow within
a watershed during dry periods. As such, the behavior of unconfined groundwater flow is key to
the physically-based understanding of the long memory in watershed runoff. Meanwhile, as will
be seen in the following derivation of its governing equation, unconfined aquifer groundwater flow
is uniquely different from the confined aquifer groundwater flow. The fundamental differences
between the two aquifer flows is that while the flow in a confined aquifer is linear and
compressible, the flow in an unconfined aquifer is nonlinear and incompressible due to the
unconfined aquifer being phreatic, its top surface boundary being open to the atmosphere.
Accordingly, hydrologists have developed unique governing equations of unconfined aquifer
groundwater flow (Bear, 1979; Freeze and Cherry, 1979). Starting with the next section, first the
continuity equation of transient unconfined groundwater flow within an anisotropic heterogeneous
aquifer under a time-space varying sink/source will be developed in fractional time and fractional
space. Then, this fractional continuity equation will be combined with a fractional groundwater
motion equation to obtain a transient groundwater flow equation in fractional time-multifractional
space within an anisotropic, heterogeneous unconfined aquifer.
Analogous to the traditional governing groundwater flow equations, as outlined by Freeze
and Cherry (1979) and Bear (1979), the fractional unconfined groundwater flow equations must
have specific features (Kavvas et al., 2017a):
i. In order for the governing equation to be prognostic, the form of the equation must be known
completely from the outset.
ii. The fractional governing equations must be dimensionally consistent and be purely
differential equations, containing only differential operators without difference operators.
iii. As the fractional derivative powers go to integer values, the fractional unconfined
groundwater flow equations must converge to the corresponding conventional integer-order
governing equations.
Within this framework, the governing equations of unconfined groundwater flow in fractional
time and fractional space will be developed in the following.
**2. Derivation of the Continuity Equation for Transient Unconfined Groundwater Flow in a**
**Heterogeneous Anisotropic Multi-Fractional Medium in Fractional Time**
To $\beta$-order the Caputo fractional derivative $D_a^{k\beta}f(x)$ of a function $f(x)$ may be defined as
( Odibat and Shawagfeh, 2007; Podlubny, 1998; Usero, 2007, Li et al., 2009),

$$D_a^{\beta}f(x) = \frac{1}{\Gamma(1-\beta)}\int_a^x \frac{f^{`}(\xi)}{(x-\xi)^{\beta}}d\xi \qquad 0<\beta<1, \quad x \geq a \quad . \tag{1}$$
where $\xi$ represents a dummy variable in the equation.
It was shown in Kavvas et al. (2017b) that one can obtain a $\beta_{x_i}$-order approximation (i=1,2)
to a function $f(x_i)$ around $x_i - \Delta x_i$ as

$$f(x_i) = f(x_i - \Delta x_i) + \frac{(\Delta x_i)^{\beta_{x_i}}}{\Gamma(\beta_{x_i}+1)} D_{x_i-\Delta x_i}^{\beta_{x_i}}f(x_i) \quad ; i=1,2. \tag{2}$$
In Equation (2), an analytical relationship between $\Delta x_i$ and $(\Delta x_i)^{\beta_{x_i}}$ (i=1,2) that will be universally
applicable throughout the modelling domain is possible when the lower limit in the above Caputo
derivative in equation (2) is taken as zero (that is, $\Delta x_i = x_i$) for f($x_i$) =$x_i$ (Kavvas et al. 2017b).
Under the Dupuit approximation of horizontal flow streamlines (very small water table
gradient) (Bear, 1979), the net mass flux through the control volume of an unconfined aquifer with
a flat bottom confining layer, as depicted in Figure 1, that also has a sink/source mass flux
$\rho q_v \Delta x_1 \Delta x_2$, can be formulated as

$\left[\rho Q_{x_1}(x_1, x_2; t) - \rho Q_{x_1}(x_1 - \Delta x_1, x_2; t)\right]\Delta x_2 + \left[\rho Q_{x_2}(x_1, x_2; t) - \rho Q_{x_2}(x_1, x_2 - \right.$
$\left. \Delta x_2; t)\right]\Delta x_1 - \rho q_v \Delta x_1 \Delta x_2$                                      (3)

where $Q_{x_i}$ is the discharge across a vertical plane of unit width in i-th direction, i = 1,2, $\rho$ is the
fluid density, and $q_v$ is the source/sink (recharge/leakage) per unit horizontal area. Then by
combining equation (2) with equation (3) with $\Delta x_i = x_i$ (i=1,2), and expressing the resulting
Caputo derivative $D_0^{\beta_{x_i}} f(x_i)$ by $\frac{\partial^{\beta_{x_i}} f(x_i)}{(\partial x_i)^{\beta_{x_i}}}$ , (i=1,2) for convenience, yields the net mass flux
through the control volume in Figure 1 to the orders of $(\Delta x_1)^{\beta_{x_1}}$ and $(\Delta x_2)^{\beta_{x_2}}$, as

$$\frac{1}{\Gamma(\beta_{x_1}+1)}\left(\frac{\partial}{\partial x_1}\right)^{\beta_{x_1}}\left(\rho Q_{x_1}(x_1, x_2; t)\right)(\Delta x_1)^{\beta_{x_1}}\Delta x_2 +$$

$$\frac{1}{\Gamma(\beta_{x_2}+1)}\left(\frac{\partial}{\partial x_2}\right)^{\beta_{x_2}}\left(\rho Q_{x_2}(x_1, x_2; t)\right)\Delta x_1 (\Delta x_2)^{\beta_{x_2}} - \rho q_v \Delta x_1 \Delta x_2 \qquad (4)$$

where different powers for fractional space derivatives are utilized in different directions due to
the anisotropy in the flow medium.

Kavvas et al. (2017b) have shown that to $\beta_{x_i}$-order fractional increments in space in the i-th
direction, i=1,2,

$$(\Delta x_i)^{\beta_{x_i}} = \frac{\Gamma(\beta_{x_i}+1)\Gamma(2-\beta_{x_i})}{x_i^{1-\beta_{x_i}}}\Delta x_i \qquad , \text{ i=1,2.} \qquad (5)$$

Combining equations (5) and (4) yields for the net mass outflow through the control volume
in Figure 1 as (to the order of $(\Delta x_i)^{\beta_{x_i}}$ , i=1,2),

$$\frac{\Gamma(2-\beta_{x_1})}{x_1^{1-\beta_{x_1}}}\left(\frac{\partial}{\partial x_1}\right)^{\beta_{x_1}}\left(\rho Q_{x_1}(\bar{x}; t)\right)\Delta x_1 \Delta x_2 + \qquad (6)$$

$$\frac{\Gamma(2-\beta_{x_2})}{x_2^{1-\beta_{x_2}}}\left(\frac{\partial}{\partial x_2}\right)^{\beta_{x_2}}\left(\rho Q_{x_2}(\bar{x}; t)\right)\Delta x_1 \Delta x_2 - \rho q_v \Delta x_1 \Delta x_2, \quad \bar{x} = (x_1, x_2).$$

Denoting the water volume within the control volume in Figure 1 by $V_w$ and using the concept
of specific yield (effective porosity) $S_y$ of a phreatic aquifer (Bear and Verruijt, 1987)
$S_y = \frac{\Delta V_w}{\Delta h}\frac{1}{\Delta x_1 \Delta x_2}$                  ,                                   (7)
where $\Delta V_w$ is the change in water volume in the control volume per change $\Delta h$ in the hydraulic
head (the elevation of the phreatic surface (water table) above the flat bottom of the aquifer ), the
time rate of change of mass within the control volume in Figure 1 may be written as (Bear and
Verruijt, 1987)

$$\frac{S_y\big(\rho h(\bar{x};t)-\rho h(\bar{x};t-\Delta t)\big)}{\Delta t}\,\Delta x_1 \Delta x_2 \tag{8}$$

which can then be expressed in terms of the approximation (2) with respect to the time dimension
as,

$\dfrac{S_y}{\Delta t}\left[\dfrac{\Delta t^{\alpha}}{\Gamma(\alpha+1)}\left(\dfrac{\partial}{\partial t}\right)^{\alpha}(\rho h)\right]\Delta x_1 \Delta x_2 \qquad \cdot$ $\tag{9}$

To $\alpha$-order fractional increments in time (Kavvas et al. 2017b)

$$(\Delta t)^{\alpha} = \frac{\Gamma(\alpha+1)\Gamma(2-\alpha)}{t^{1-\alpha}}\,\Delta t \qquad \cdot \tag{10}$$

Substituting equation (10) into equation (9), one can obtain the time rate of change of mass in the
control volume, as shown in Figure 1;

$$S_y \frac{\Gamma(2-\alpha)}{t^{1-\alpha}}\left(\frac{\partial}{\partial t}\right)^{\alpha}(\rho h)\,\Delta x_1 \Delta x_2 \;. \tag{11}$$


As the time rate of change of mass within the control volume, as shown in Figure 1, must be
inversely proportional to the net mass flux passing through the control volume, one may combine
equations (6) and (11) to obtain

$\left[\dfrac{\Gamma(2-\beta_{x_1})}{x_1^{1-\beta_{x_1}}}\left(\dfrac{\partial}{\partial x_1}\right)^{\beta_{x_1}}\left(\rho Q_{x_1}(\bar{x};t)\right)+\dfrac{\Gamma(2-\beta_{x_2})}{x_2^{1-\beta_{x_2}}}\left(\dfrac{\partial}{\partial x_2}\right)^{\beta_{x_2}}\left(\rho Q_{x_2}(\bar{x};t)\right)-\rho q_v\right]\Delta x_1 \Delta x_2 =$
$\qquad -S_y\dfrac{\Gamma(2-\alpha)}{t^{1-\alpha}}\left(\dfrac{\partial}{\partial t}\right)^{\alpha}(\rho h)\,\Delta x_1 \Delta x_2$ $\tag{12}$

$\dfrac{\Gamma(2-\beta_{x_1})}{x_1^{1-\beta_{x_1}}}\left(\dfrac{\partial}{\partial x_1}\right)^{\beta_{x_1}}\left(\rho Q_{x_1}(\bar{x};t)\right)+\dfrac{\Gamma(2-\beta_{x_2})}{x_2^{1-\beta_{x_2}}}\left(\dfrac{\partial}{\partial x_2}\right)^{\beta_{x_2}}\left(\rho Q_{x_2}(\bar{x};t)\right)-\rho q_v = -S_y\dfrac{\Gamma(2-\alpha)}{t^{1-\alpha}}\left(\dfrac{\partial}{\partial t}\right)^{\alpha}(\rho h)$ $\tag{13}$
for $0 < \alpha, \beta_{x_1}, \beta_{x_2} < 1$, $\bar{x} = (x_1, x_2,)$.
Within the framework of fluid incompressibility in the unconfined aquifer, equation (13)
reduces further to

$$\frac{\Gamma(2-\beta_{x_1})}{x_1^{1-\beta_{x_1}}} \left(\frac{\partial}{\partial x_1}\right)^{\beta_{x_1}} \left(Q_{x_1}(\bar{x};t)\right) + \frac{\Gamma(2-\beta_{x_2})}{x_2^{1-\beta_{x_2}}} \left(\frac{\partial}{\partial x_2}\right)^{\beta_{x_2}} \left(Q_{x_2}(\bar{x};t)\right) - q_v = -S_y \frac{\Gamma(2-\alpha)}{t^{1-\alpha}} \frac{\partial^\alpha h}{(\partial t)^\alpha}$$

$$\frac{\Gamma(2-\beta_{x_1})}{\Gamma(2-\alpha)} \frac{t^{1-\alpha}}{x_1^{1-\beta_{x_1}}} \left(\frac{\partial}{\partial x_1}\right)^{\beta_{x_1}} \left(Q_{x_1}(\bar{x};t)\right) + \frac{\Gamma(2-\beta_{x_2})}{\Gamma(2-\alpha)} \frac{t^{1-\alpha}}{x_2^{1-\beta_{x_2}}} \left(\frac{\partial}{\partial x_2}\right)^{\beta_{x_2}} \left(Q_{x_2}(\bar{x};t)\right) - \frac{t^{1-\alpha}}{\Gamma(2-\alpha)} q_v \rightarrow$$
$$= -S_y \frac{\partial^\alpha h}{(\partial t)^\alpha}$$

(14)

for $0 < \alpha, \beta_{x_1}, \beta_{x_2} < 1$, $\bar{x} = (x_1, x_2,)$ as the time-space fractional continuity equation of transient
groundwater flow in an anisotropic unconfined aquifer with multi-fractional dimensions and in
fractional time.
Performing a dimensional analysis of equation (14) yields

$$\frac{L}{T^\alpha} = \frac{T^{1-\alpha}}{L^{1-\beta_{x_1}}} \frac{1}{L^{\beta_{x_1}}} \frac{L^2}{T} = \frac{T^{1-\alpha}}{L^{1-\beta_{x_2}}} \frac{1}{L^{\beta_{x_2}}} \frac{L^2}{T} = \frac{T^{1-\alpha}}{1} \frac{L}{T} = \frac{L}{T^\alpha}$$     (15)

where L denotes length and T denotes time. Also, $\alpha, \beta_{x_1}$ and $\beta_{x_2}$ are respectively the fractional
powers in time and $x_1$ and $x_2$ spatial dimensions. In equation (15), starting from the left-hand-side
(LHS), the first term shows the final dimension of equation (14), the second term shows in detail
the dimensions of the individual components of the first term on the LHS of equation (14), the
third term shows in detail the dimensions of the individual components of the second term on the
LHS of equation (14), the fourth term shows in detail the dimensions of the individual components
of the third term on the LHS of equation (14), and the fifth and the last term shows in detail the
dimensions of the individual components on the right-hand-side (RHS) of equation (14). Hence,
the left-hand and right-hand sides of the continuity equation (14) for transient groundwater flow
in an unconfined aquifer in multi-fractional space and fractional time are consistent as shown in
equation (15).
For $n\text{-}1 < \alpha, \beta_{x_i} < n$ where n is any positive integer, as $\alpha$ and $\beta_{x_i} \rightarrow$ n, the Caputo fractional
derivative of a function f(y) to order $\alpha$ or $\beta_{x_i}$ (i = 1, 2) yields the standard n-th derivative of the
function f(y) (Podlubny, 1998). Then when $\alpha$ and $\beta_{x_i} \rightarrow 1$ (i = 1, 2), the continuity equation (14)
becomes the conventional continuity equation for transient groundwater flow in an unconfined
aquifer:

$$-S_y \frac{\partial h}{\partial t} = \frac{\partial}{\partial x_1}\left(Q_{x_1}(\bar{x}; t)\right) + \frac{\partial}{\partial x_2}\left(Q_{x_2}(\bar{x}; t)\right) - q_v \ . \tag{16}$$

## 3. Motion Equation (Specific Discharge Equation) in Fractional Multi-Dimensional Unconfined Aquifers

Recently, Kavvas et al., (2017a, 2017b) derived a governing equation for water flux
(specific discharge), $q_{x_i}$, (i = 1, 2, 3) in a saturated or unsaturated porous medium with fractional
dimensions in the form,

$$q_i(\bar{x}, t) = -K_{s,x_i}(\bar{x}) \frac{\Gamma(2-\beta_{x_i})}{x_i^{1-\beta_{x_i}}} \frac{\partial^{\beta_{x_i}} h}{(\partial x_i)^{\beta_{x_i}}}, \ i = 1,2,3; \ \ \bar{x} = (x_1, x_2, x_3). \tag{17}$$

where $K_{s,x_i}(\bar{x})$ is the saturated hydraulic conductivity in the i-th spatial direction (i=1,2,3).
Meanwhile, under the Dupuit approximation of essentially horizontal unconfined aquifer flow
(water table slope very small) (Bear, 1979), referring to Figure 1, the discharge per unit width in
the i-th direction (i = 1,2) can be expressed as
$$Q_{x_i}(\bar{x}, t) = h q_i(\bar{x}, t), \ i = 1,2 \ \ \ ; \ \ \bar{x} = (x_1, x_2,). \tag{18}$$
Then combining equations (18) and (17) results in
$$Q_{x_i}(\bar{x}, t) = -K_{s,x_i}(\bar{x}) \frac{\Gamma(2-\beta_{x_i})}{x_i^{1-\beta_{x_i}}} h \frac{\partial^{\beta_{x_i}} h}{(\partial x_i)^{\beta_{x_i}}} \ \ , i = 1,2; \ \bar{x} = (x_1, x_2,) \tag{19}$$
as the governing equation of groundwater motion within an unconfined aquifer with a flat bottom
confining layer. In equation (19) h is the unconfined aquifer thickness or the phreatic surface
elevation above the bottom confining layer.
A dimensional analysis on equation (19) yields $L^2/T$ for the units of both the left-hand-side
(LHS) and the RHS of the equation, establishing its dimensional consistency.
Applying the above-mentioned result of Podlubny (1998) on the convergence of a fractional
derivative to a corresponding integer derivative for $\beta_{x_i} \to 1$ (i = 1, 2) reduces the fractional motion
equation (19) for unconfined groundwater flow to the conventional equation (Bear, 1979):

$$Q_{x_i}(\bar{x}, t) = -K_{s,x_i}(\bar{x})h\frac{\partial h(\bar{x},t)}{\partial x_i}, \text{ i= 1,2} \tag{20}$$

for the case of integer spatial dimensions. As such, the fractional motion equation (19) for
unconfined groundwater flow in fractional spatial dimensions is consistent with the conventional
motion equation for the integer spatial dimensions.
**4. The Complete Equation for Transient Unconfined Groundwater Flow in Multi-Fractional**
**Space and Fractional Time**
Combining the fractional motion equation (19) of groundwater flow in an unconfined aquifer
with the fractional continuity equation (14) of unconfined groundwater flow results in the equation,

$$S_y \frac{\partial^\alpha h}{(\partial t)^\alpha} = \frac{\Gamma(2-\beta_{x_1})}{x_1^{1-\beta_{x_1}}} \left(\frac{\partial}{\partial x_1}\right)^{\beta_{x_1}} \left(K_{s,x_1}(\bar{x}) \frac{t^{1-\alpha}}{x_1^{1-\beta_{x_1}}} \frac{\Gamma(2-\beta_{x_1})}{\Gamma(2-\alpha)} h \frac{\partial^{\beta_{x_1}} h}{(\partial x_1)^{\beta_{x_1}}}\right) +$$
$$\frac{\Gamma(2-\beta_{x_2})}{x_2^{1-\beta_{x_2}}} \left(\frac{\partial}{\partial x_2}\right)^{\beta_{x_2}} \left(K_{s,x_2}(\bar{x}) \frac{t^{1-\alpha}}{x_2^{1-\beta_{x_2}}} \frac{\Gamma(2-\beta_{x_2})}{\Gamma(2-\alpha)} h \frac{\partial^{\beta_{x_2}} h}{(\partial x_2)^{\beta_{x_2}}}\right) + \frac{t^{1-\alpha}}{\Gamma(2-\alpha)} q_v \tag{21}$$

for $0 < \alpha, \beta_{x_1}, \beta_{x_2} < 1$, $\bar{x} = (x_1, x_2,)$ as the time-space fractional governing equation of transient
unconfined groundwater flow in an anisotropic medium.
Performing a dimensional analysis of Equation (21) yields

$$\frac{L}{T^\alpha} = \frac{1}{L^{1-\beta_{x_1}}} \frac{1}{L^{\beta_{x_1}}} \frac{L}{T} \frac{T^{1-\alpha}}{L^{1-\beta_{x_1}}} L \frac{L}{L^{\beta_{x_1}}} = \frac{1}{L^{1-\beta_{x_2}}} \frac{1}{L^{\beta_{x_2}}} \frac{L}{T} \frac{T^{1-\alpha}}{L^{1-\beta_{x_2}}} \frac{L^2}{L^{\beta_{x_2}}} = \frac{T^{1-\alpha}}{1} \frac{L}{T} = \frac{L}{T^\alpha} \tag{22}$$

where L denotes length and T denotes time. Hence, the left-hand and right-hand sides of the
governing equation (21) for transient groundwater flow in an unconfined aquifer in multi-
fractional space and fractional time are consistent.
Specializing the above-discussed result of Podlubny (1998) to n = 1, for $\alpha$ and $\beta_{x_i} \to 1$ ( i =
1, 2) reduces the governing fractional equation (21) to the conventional governing equation for
transient groundwater flow in an unconfined aquifer (Bear, 1979):
$$S_y \frac{\partial h}{\partial t} = \frac{\partial}{\partial x_1}\left(K_{s,x_1}(\bar{x})h\frac{\partial h(\bar{x},t)}{\partial x_1}\right) + \frac{\partial}{\partial x_2}\left(K_{s,x_2}(\bar{x})h\frac{\partial h(\bar{x},t)}{\partial x_2}\right) + q_v \qquad (23)$$

## 5. Numerical application


To demonstrate the skills of the proposed fractional governing equation of unconfined aquifer
groundwater flow, two numerical applications are performed using the proposed fractional
governing equation. The first application follows the physical setting of an example from Wang
and Anderson (1995), as depicted in Figure 2. The numerical problem of seepage through a dam
under a sudden change in the water surface elevation at the downstream section of the dam is
modified based on seepage through a dam, Page 53 and Problem 4.4 (a), Page 89 in Wang and
Anderson (1995), as shown in Figure 2. The water seepage through the dam's body may be
interpreted as one-dimensional groundwater flow through an unconfined aquifer. The unconfined
flow system locates the top boundary of the saturated zone in an earthen dam and the bottom of
the dam rests on impermeable rock. For this example, the unconfined aquifer length L is 100 m.
The initial water level in the upstream and downstream sections of the dam and through the dam's
body is 16 m. Then immediately after the initial time, the water level at the downstream section of
the dam is suddenly dropped to 11 m and remains as 11 m afterwards. The unconfined aquifer
parameters are $S = 0.2$ for the specific yield and $K=0.002$ m/min for the hydraulic conductivity,
respectively. The analytical solution for this problem at the steady-state is:
$$h = \sqrt{\frac{h_2^2 - h_1^2}{L}x + h_1^2} \qquad (24)$$
where $h$ is the depth of the unconfined groundwater surface from the bottom layer; $L$ is the aquifer
length; $x$ is the distance from the upstream location of the dam body, and $h_1$ and $h_2$ are as shown
in Figure 2.
In Figures 3(a) and 3(b), the normalized groundwater head and normalized groundwater
discharge per unit width at location $x=L/2$ through time under different fractional power values
are shown. Meanwhile, Figure 3(c) shows the normalized groundwater head at the time instance
t=40,000 min as a function of location throughout the dam's body, and the analytical solution of
the standard governing equation of unconfined groundwater flow when $\beta_x = \alpha = 1$ at the steady
state. The considered fractional derivative powers in space and time are $\beta_x = \alpha =$
0.7, 0.8, 0.9, 1.0. As can be seen from Figure 3(a), the hydraulic head recession in time slows down
with the decrease of $\beta_x = \alpha$ from 1. The hydraulic heads in Figure 3(a) have heavier tails as orders
of time and space fractional derivative powers decrease from 1 towards 0.7. Furthermore,
normalized groundwater discharge per unit width in Figure 3(b) goes to 1 in a slower rate as
fractional derivative powers decrease from 1 towards 0.7. Meanwhile, Figure 3(c) shows that the
numerical solution of the governing fractional equation at $\beta_x = \alpha = 1.0$ and at a very long time
after the initial condition, matches perfectly the steady state analytical solution (24) of the standard
equation (23) with the specified initial/boundary conditions.

The second application deals with a transient unconfined groundwater flow from a hillslope

toward a stream (Figure 4). The upstream boundary plane separates the region of flow from the
adjacent hillslope that feeds the adjacent tributary system, therefore $\frac{\partial h}{\partial x} = 0$ (Freeze, 1978) at x=0.
The normalized initial groundwater head in the unconfined aquifer, and the normalized
groundwater head at time t=60,000 min through the length of the aquifer under different fractional
derivative powers are shown in Figure 5(a). The normalized groundwater head and normalized
groundwater discharge per unit width at *x=L/2* through time under different fractional derivative
powers are demonstrated in Figures 5(b) and 5(c). As can be seen from Figures 5(b)-(c), the
hydraulic head and groundwater discharge recession in time slows down with the decrease of $\beta_x =$
$\alpha$ from 1. The hydraulic heads and groundwater discharges in Figures 5(b)-(c) have heavier tails
as orders of time and space fractional derivative powers decrease from 1 towards 0.7.
**6. Discussion**

From the standard governing equation (23) of unconfined groundwater flow in integer time-

space the saturated hydraulic conductivity may be interpreted as a diffusion coefficient for the
nonlinear diffusion of groundwater in an unconfined aquifer. The basic difference between
confined and unconfined groundwater flow is that the former can be interpreted as a linear
diffusion of groundwater while the latter is a nonlinear diffusion of groundwater within an
aquifer. Similar to saturated hydraulic conductivities in equation (26) in Kavvas et al., (2017a)
for the fractional confined aquifer groundwater flow, the saturated hydraulic conductivities in
equation (21) above, which governs fractional unconfined aquifer groundwater flow, are
modulated by the ratios of fractional time to fractional space, $\frac{t^{1-\alpha}}{x_i^{1-\beta x_i}}$ , i= 1,2. In other words, the
confined and unconfined groundwater diffusions in fractional time-space are modulated by the
above fractional time-space ratios.
Numerical application demonstrated that as the powers of the space and time fractional
derivatives decrease from 1, the recession rate of the nondimensional groundwater hydraulic
head slows down when compared to the case by the conventional governing equation (i.e., with
integer order derivatives). This behavior also indicates the modulation of the nonlinear diffusion
of the groundwater by the fractional powers of time and space.
As mentioned in the Introduction section, unconfined groundwater flow is the fundamental
component of the watershed runoff baseflow since it is the fundamental contributor to the
streamflow network within a watershed during dry periods. As such, the behavior of unconfined
groundwater flow is key to the physically-based understanding of the long memory in watershed
runoff. As seen from the numerical applications in Figures 3 and 5, the powers of the fractional
derivatives in time and space can modulate the speed of the groundwater discharge evolution.
Hence, they can modulate the memory of the unconfined aquifer flow, which, in turn, can modulate
the memory of the watershed baseflow. Meanwhile, the Caputo derivative, as defined in its special
form $D_0^{\beta_{x_i}} f(x_i)$ in space in this study, was shown by Kavvas and Ercan (2017) and Ercan and
Kavvas (2017) to be a nonlocal quantity where the effect of the boundary conditions on the
groundwater flow within the flow domain can have long spatial memories with the decrease in the
powers of the spatial fractional derivatives from unity. Similarly, it was shown by Kavvas et al.
(2017a) that the Caputo derivative in time, $D_0^{\alpha} f(t)$, as defined in this study, is nonlocal in time,
and can carry the effect of initial conditions on the groundwater flow for long times as the power
in the time fractional derivative decreases from 1. Therefore, the fractional governing equation of
unconfined groundwater flow in fractional time and multi-fractional space has the potential to
describe the long memory characteristics of baseflow within a watershed. This important topic
shall be explored in the near future.

7. **Conclusion**

A dimensionally-consistent fractional governing equation of transient unconfined aquifer
groundwater flow was derived within fractional differentiation framework. After developing a
fractional continuity equation, a previously-developed dimensionally consistent equation for water
flux in unsaturated/saturated porous media was combined with the Dupuit approximation to obtain
an equation for groundwater motion in multi-fractional space in unconfined aquifers. Combining
the fractional continuity and motion equations, the governing equation of transient unconfined
aquifer groundwater flow in a multi-fractional medium in fractional time was then obtained. To
demonstrate the skills of the proposed fractional governing equation of unconfined aquifer
groundwater flow, two numerical applications were performed. As demonstrated in the numerical
application results, the orders of the fractional space and time derivatives modulate the speed of
groundwater discharge and groundwater flow evolution, slowing the process with decrease in the
powers of the fractional derivatives from 1. It is also shown that the proposed dimensionally
consistent fractional governing equations approach to the corresponding conventional equations
as the fractional orders of the derivatives go to 1.

**Data availability.**
The data used in this article can be accessed by contacting the corresponding author.
**Appendix A. Numerical Solution for 1-dimensional case**
One-dimensional time-space fractional groundwater flow in the unconfined aquifer with no
recharge or leakage can be written as:
$$S_y \frac{\partial^\alpha h}{(\partial t)^\alpha} = \frac{\Gamma(2-\beta)}{x^{1-\beta}} \left(\frac{\partial}{\partial x}\right)^\beta \left(K_s(\bar{X}) \frac{t^{1-\alpha}}{x^{1-\beta}} \frac{\Gamma(2-\beta)}{\Gamma(2-\alpha)} h \frac{\partial^\beta h}{(\partial x)^\beta}\right) \tag{A1}$$

The fractional time and space derivatives are estimated in the same manner as that in Tu et al.
(2018), where the Caputo fractional space and time derivatives in the fractional governing equation
are estimated by the numerical algorithm in Odibat (2009) and the algorithm reported by Murio
(2008), respectively. The Caputo fractional space derivative $D_x^\beta g(x)\big|_{x=L}$ at the location $L$ for $m$-1 <
$\beta \leq m$ ( $m \bar{\mathsf{l}}\ N$ ) of a given space interval [0, $L$] is estimated as:

$$D_x^b g(x)\big|_{x=L} \approx \frac{DL^{m-b}}{G(m+2-b)} \left\{ \left[ (N-1)^{m-b+1} - (N-m+b-1)DL^{m-b}\right] g^{(m)}(0) + g^{(m)}(L) \right.$$
$$\left. + \sum_{i=1}^{N-1} \left[ (N-i+1)^{m-b+1} - 2(N-i)^{m-b+1} + (N-i-1)^{m-b+1}\right] g^{(m)}(l_i) \right\} \tag{A2}$$

where $N$ is the number of equally spaced subintervals on [0, $L$]; the subinterval length is $\Delta L = L/N$,
and $l_i = i\Delta L$, for $i = 0,1,2,\ldots,N$.
The Caputo fractional time derivative $D_t^a g(x,t)\big|_{x=l_i,t=t_n}$ for $0 < \alpha \leq 1$ on a given time interval
[0, T], which is divided into M equal subintervals with a time window of $\Delta t = T/M$ by using the
nodes $t_n = n\Delta t$, $n = 0, 1, 2,..., M$, can be approximated as:

$$D_t^a g_i^n = \frac{\mathrm{D}t^{-a}}{\mathrm{G}(2-a)} \sum_{k=1}^{n} \left[ k^{1-a} - (k-1)^{1-a} \right] \left( g_i^{n-k+1} - g_i^{n-k} \right) \tag{A3}$$

Then the 1-D governing equation in fractional time and space for Cartesian groundwater flow in
an unconfined aquifer can be discretized as:
For $n = 1$,

$$h_i^n = h_i^{n-1} + \frac{t_n^{1-a}}{S_y \mathrm{D}t^{-a}} \frac{\mathrm{G}(2-b)}{l_i^{1-b}} G_i^{n-1} \tag{A4}$$

For $n \geq 2$,

$$h_i^n = h_i^{n-1} + \frac{t_n^{1-a}}{S_y \mathrm{D}t^{-a}} \frac{\mathrm{G}(2-b)}{l_i^{1-b}} G_i^{n-1} - \sum_{k=2}^{n} \left[ k^{1-a} - (k-1)^{1-a} \right] \left( h_i^{n-k+1} - h_i^{n-k} \right) \tag{A5}$$

where $G = \left( \dfrac{\partial}{\partial x} \right)^b \left[ K_s(\overline{X}) \dfrac{\mathrm{G}(2-b)}{x^{1-b}} h \dfrac{\partial^b h}{\partial x^b} \right]$ and the space and time fractional
derivatives in G are estimated as in Equations (A2) and (A3).
**Competing interests.**
The authors declare that they have no conflict of interest.

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

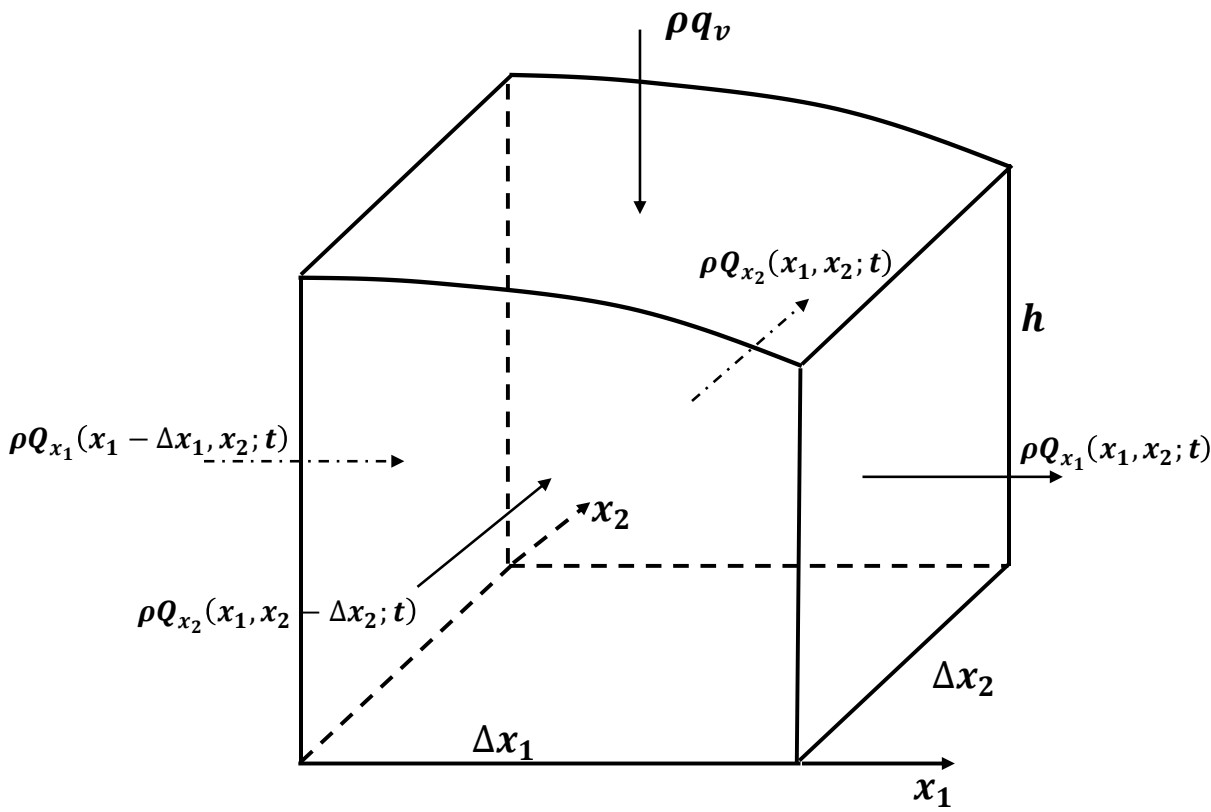


Figure 1. The mass flux through the control volume of an unconfined aquifer.



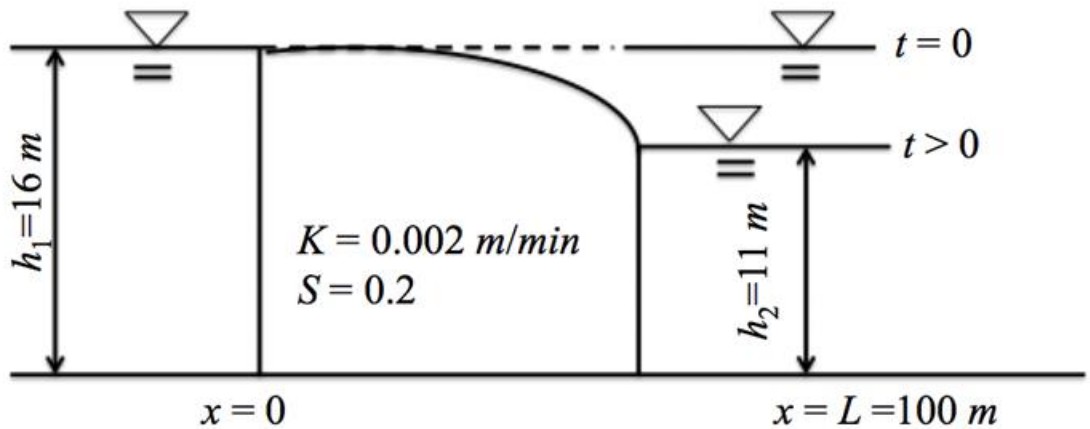


Figure 2. The sketch of numerical application 1: Water seepage through a dam's body as an
unconfined groundwater flow

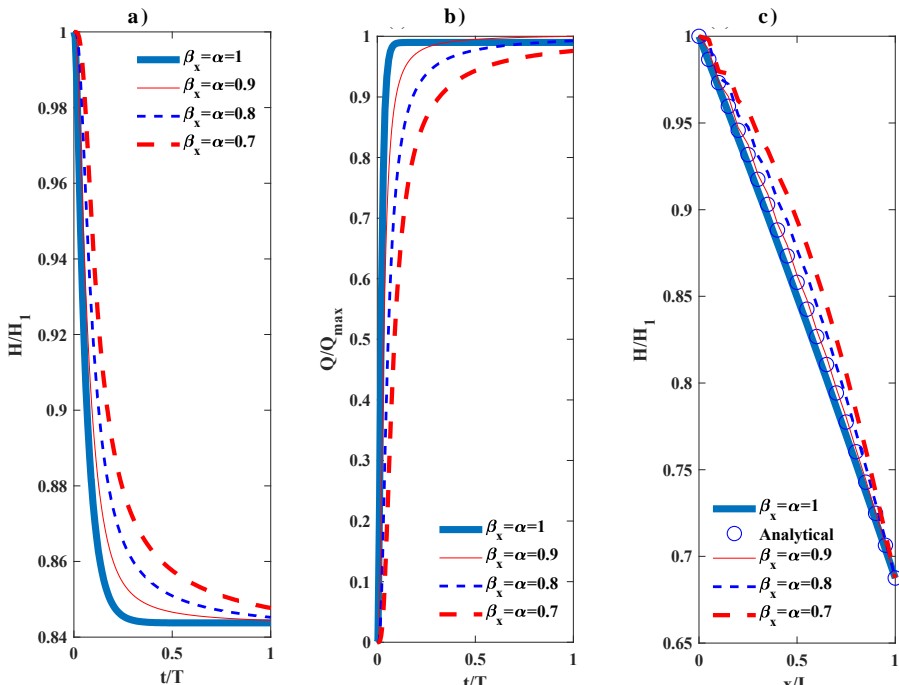

Figure 3. Results for numerical application 1: (a) The normalized groundwater head at *x=L/2*
through time under different fractional derivative powers; (b) The normalized groundwater
discharge per unit width at *x=L/2* through time under different fractional derivative powers; *t* is
time and the simulation time *T* is 120,000 min; (c) The normalized groundwater head at t=40,000
min through length of the aquifer (through the body of the dam) and the analytical solution of the
conventional governing equation of unconfined groundwater flow when $\beta_x = \alpha = 1$ at the steady
state.

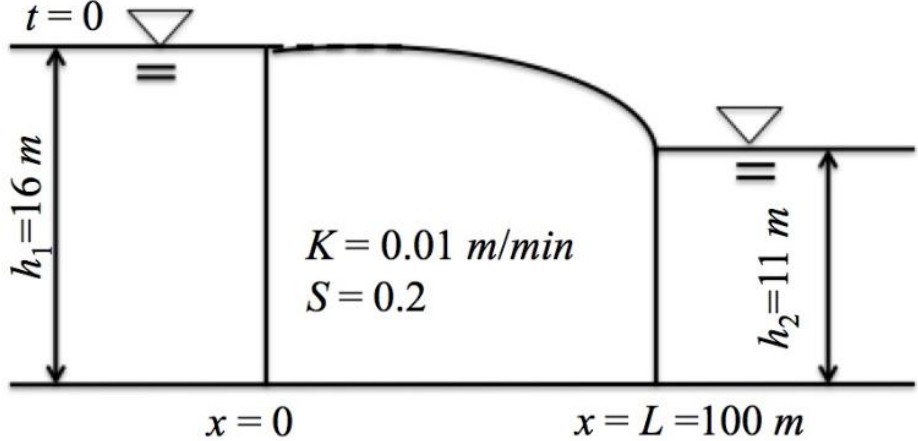

Figure 4. The sketch of numerical application 2: The downstream groundwater head is fixed at 11
m and the initial upstream groundwater head is 16 m.


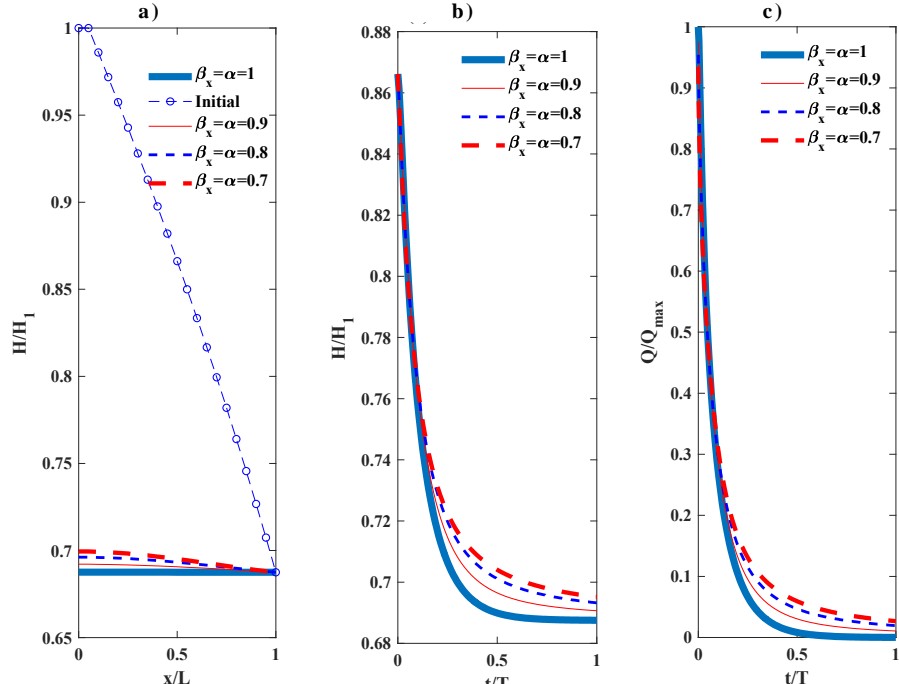

Figure 5. Results for numerical application 2: (a) The normalized initial groundwater head in the
unconfined aquifer, and the normalized groundwater head at time t=60,000 min through length of
the aquifer under different fractional derivative powers; (b) The normalized groundwater head at
*x=L*/2 through time under different fractional derivative powers; (c) The normalized groundwater
discharge per unit width at *x=L*/2 through time under different fractional derivative powers; *t* is
time and the simulation time *T* is 60,000 min.

