# Peer review of "Fractional governing equations of transient groundwater flow in unconfined"

_Earth System Dynamics, 2019_

## Referee Comment (RC1) · Anonymous Referee #1 · 25 Aug 2019

General comments:

This paper deals with the theoretical study of deriving the governing equation of unconfined aquifer flow using Caputo fractional derivative approach. The derivation process is very clearly presented for the reader to understand. Including further discussion of the following is expected to further enhance the value of the fundamental research work.

Specific comments:

[Figure]

1) The paper needs to contain the minimum information of the numerical scheme needed to draw Figure 3. This will provide important information to persuade the paper's reproducibility. 2) The authors simulated a state after a very long time to draw Figure 3 (b). For integer cases, one can derive a simple steady-state analytical solution, as shown in Eq. 24. However, This reviewer is curious about what the fractional case might look like. It is necessary to include the authors' views on this curiosity. 3) Further discussion is needed about the time required to converge to a steady state. The time required will naturally be affected by the fractional order. 4) In addition to the head results, the authors need to explain the behavior of the discharge. In the case of integer cases, the discharge at steady-state can be derived analytically simply, but what happens in the case of fractional cases, and the effects of fractional order on steady-state discharge need to be discussed further.

Technical corrections:

Too many terms are given in eq. (15). Matching the order and number of terms in eqs. (14) and (15) will help readers better understand. Line 255: storage coefficient S = 0.2 popped out abruptly without any explanation, and the effective porosity Sy is missing a description of what value is given in the numerical analysis.

---

## Referee Comment (RC2) · Anonymous Referee #2 · 30 Aug 2019

It will be nice to have a global approach that has some background and reason for the research in the first paragraph of the introduction. Right now, it is jumping to the problem. The introduction section has adequate literature reviewed to come to the present research. Such is good but there are numerous jargon to be defined or clearly mentioned. For example, Riemann-Liouville fractional derivative, local Caputo derivatives. The text explained the intensive use of such derivatives. But for the general audience, the questions could arise how such derivatives were used. What could be the assumptions? The reviewer suggests revising the introduction section with implicit

assumptions behind them.

It seems that the authors tried to stick to a book chapter. It is not clear why this problem is strictly considered. Since this is the research paper, one should try with a real problem, not the virtual ones. The conclusion made by the authors is too technical. The reviewer does not see any possible application as well as future research behind this.

In equation (1) the definition of shi is strictly missing. In line 168, a comma is extra. In lines 286-287, the phrase "the network streamflow" should be better if it is like "the streamflow network."

---

## Author Comment (AC1) · 8 Oct 2019

**Response to Comments of Anonymous Referee #1**

Authors thank Reviewer #1 for the valuable comments and suggestions, which helped to improve the manuscript considerably. Our responses are in blue color below.

**Referee #1:** General comments:
This paper deals with the theoretical study of deriving the governing equation of unconfined aquifer flow using Caputo fractional derivative approach. The derivation process is very clearly presented for the reader to understand. Including further discussion of the following is expected to further enhance the value of the fundamental research work.
**Authors' response:** Thank you, please see our responses below.

**Referee #1:** Specific comments: 1) The paper needs to contain the minimum information of the numerical scheme needed to draw Figure 3. This will provide important information to persuade the paper's reproducibility.
**Authors' response:** Information about numerical scheme is added as an appendix to the revised manuscript.

**Referee #1:** 2) The authors simulated a state after a very long time to draw Figure 3 (b). For integer cases, one can derive a simple steady-state analytical solution, as shown in Eq. 24. However, This reviewer is curious about what the fractional case might look like. It is necessary to include the authors' views on this curiosity.
**Authors' response:** Derivation of an analytical solution for the proposed fractional governing equation of unconfined groundwater flow is not the focus of this study. Since such an analytical solution is not available currently, the authors will address this issue in future studies.

**Referee #1:** 3) Further discussion is needed about the time required to converge to a steady state. The time required will naturally be affected by the fractional order.
**Authors' response:** Further discussion is provided in the revised manuscript for two numerical applications for various fractional orders.

**Referee #1:** 4) In addition to the head results, the authors need to explain the behavior of the discharge. In the case of integer cases, the discharge at steady-state can be derived analytically simply, but what happens in the case of fractional cases, and the effects of fractional order on steady-state discharge need to be discussed further.
**Authors' response:** A figure for the discharge is added to the revised manuscript (Figure 3c, see below). A discussion of the new figure for flows by the standard integer order and fractional order governing equations are also provided in the revised manuscript. It takes longer time to achieve steady state conditions as fractional powers decrease from 1 toward zero.

[Figure]

Figure 3. Results for numerical application 1: (a) The normalized groundwater head $h/h_1$ at $x=L/2$ through time under different fractional derivative powers; (b) The normalized groundwater head $h/h_1$ at t=40,000 min through the length of the aquifer (through the body of the dam) and the analytical solution of the conventional governing equation of unconfined groundwater flow when $\beta_x = \alpha = 1$ at the steady state; (c) The normalized groundwater discharge per unit width at $x=L/2$ through time under different fractional derivative powers; $t$ is time and the simulation time $T$ is 120,000 min.

In order to further satisfy reviewer's comment on flows, we added a second numerical example. The second problem deals with a transient unconfined groundwater flow from a hillslope toward a stream (Figure 4, see below). The upstream boundary vertical plane separates the region of flow from the adjacent hillslope that feeds the adjacent tributary system, therefore $\frac{\partial h}{\partial x} = 0$ at x=0 (Freeze, 1978).

As shown in Figure 5c in the revised manuscript (see below), the newly-developed governing equations can produce heavy-tailed recession behavior in unconfined aquifer discharges by changing fractional powers.

[Figure]

Figure 4. The sketch of numerical application 2: The downstream groundwater head is fixed at 11 m and the initial upstream groundwater head is 16 m.

[Figure]

Figure 5. Results for numerical application 2: (a) The normalized groundwater head $H/H_1$ at $x=L/2$ through time under different fractional derivative powers; (b) The normalized initial groundwater head in the unconfined aquifer, and the normalized groundwater head $H/H_1$ at time $t=60,000$ min through length of the aquifer under different fractional derivative powers; (c) The normalized groundwater discharge per unit width at $x=L/2$ through time under different fractional derivative powers; $t$ is time and the simulation time $T$ is 60,000 min.

Reference
Freeze, R. A., "Mathematical models of hillslope hydrology", Chap. 6 in *Hillslope Hydrology*, ed. by Kirkby, M.J. John Wiley & Sons, Ltd, New York, 1978.

**Referee #1:** Technical corrections:
Too many terms are given in eq. (15). Matching the order and number of terms in eqs. (14) and (15) will help readers better understand.
**Authors' response:** The manuscript was revised according to the specific suggestion of the reviewer.

**Referee #1:** Technical corrections:
Line 255: storage coefficient S = 0.2 popped out abruptly without any explanation, and the effective porosity Sy is missing a description of what value is given in the numerical analysis.
**Authors' response:** The manuscript was revised according to the specific suggestions of the reviewer.

---

## Author Comment (AC2) · 8 Oct 2019

**Response to Comments of Anonymous Referee #2**

Authors thank Reviewer #2 for the valuable comments and suggestions. We believe that the manuscript is improved considerably after the suggested revisions. Our responses are in blue color below.

**Referee #2:** It will be nice to have a global approach that has some background and reason for the research in the first paragraph of the introduction. Right now, it is jumping to the problem. …
The reviewer suggests revising the introduction section with implicit assumptions behind them.
**Authors' response:** Thank you for the valuable suggestion. The Introduction section was revised to include the below section in the revised manuscript:

[revised manuscript text omitted]

**Referee#2:** The introduction section has adequate literature reviewed to come to the present research. Such is good but there are numerous jargon to be defined or clearly mentioned. For example, Riemann-Liouville fractional derivative, local Caputo derivatives. The text explained the intensive use of such derivatives. But for the general audience, the questions could arise how such derivatives were used. What could be the assumptions?

**Authors' response:** Thank you for the valuable suggestion. The Introduction section was revised to address the issues raised by the reviewer by providing detailed references to the Riemann-Liouville and Caputo fractional derivatives, and by providing the physical reasons for preferring the Caputo fractional derivative over the Riemann-Liouville derivative in this study.

**Referee #2:** It seems that the authors tried to stick to a book chapter. It is not clear why this problem is strictly considered. Since this is the research paper, one should try with a real problem, not the virtual ones. The conclusion made by the authors is too technical. The reviewer does not see any possible application as well as future research behind this.

**Authors' response:** The particular numerical example was chosen because the analytical solution of the corresponding unconfined ground water flow is available in the referenced book chapter. This problem was chosen to demonstrate that our numerical solution can reproduce the standard unconfined groundwater flow problem and how the solution varies by changing the fractional derivative powers of the proposed fractional governing equations.

In order to further satisfy the reviewer's concerns about the applicability of the research, we added a second numerical example. The second problem deals with a transient unconfined groundwater flow from

a hillslope toward a stream (Figure 4 in the revised manuscript). The upstream boundary vertical plane separates the region of flow from the adjacent hillslope that feeds the adjacent tributary system, therefore $\frac{\partial h}{\partial x} = 0$ (Freeze, 1978).

As shown in Figure 5c in the revised manuscript, the newly-developed governing equations can produce heavy-tailed recession behavior in unconfined aquifer discharges.

[Figure]

Figure 4. The sketch of numerical application 2: The downstream groundwater head is fixed at 11 m and the initial upstream groundwater head is 16 m.

[Figure]

Figure 5. Results for numerical application 2: (a) The normalized groundwater head $H/H_1$ at $x=L/2$ through time under different fractional derivative powers; (b) The normalized initial groundwater head in the unconfined aquifer, and the normalized groundwater head $H/H_1$ at time t=60,000 min through length of the aquifer under different fractional derivative powers; (c) The normalized groundwater discharge per unit width at $x=L/2$ through time under different fractional derivative powers; $t$ is time and the simulation time $T$ is 60,000 min.

**Authors' response:** Revised as suggested.

---

## Author Response (AR1)

Dear Dr. Perdigão,

Authors revised the manuscript based on the suggestions and comments of the anonymous reviewers. Our specific responses to the reviewers were provided on October 8, 2019. The revised manuscript is uploaded to the editorial system.

With Best regards,

Dr. Ali Ercan,

On behalf of the authors